# Fairtrade and Market Efficiency: Fairtrade-Labeled Coffee in the Swedish Coffee Market

**Dick Durevall**

HUI Research, Stockholm and Department of Economics, School of Business, Economics and Law, University of Gothenburg, Box 640, SE 405 30 Gothenburg, Sweden; dick.durevall@gu.se

**Abstract:** Fairtrade labeling has the potential to increase market efficiency by connecting farmers to altruistic consumers who are willing to pay a premium for sustainability-certified products. A requirement for increased efficiency, though, is that the farmers' benefits are larger than the Fairtrade processing costs and the excess payment by consumers that does not accrue to farmers; otherwise direct transfers to farmers would be more efficient. This paper analyzes how excess payment for Fairtrade-labeled coffee is distributed in the Swedish market, using information on production costs and scanner data on almost all roasted and ground coffee products sold by retailers. A key finding is that roasters and retailers get 61–70%, while producer countries, in this paper comprising coffee farmers, cooperatives, middlemen, exporters, and Fairtrade International, get 24–31%; Fairtrade Sweden gets 6–8%. These values are the upper and lower bounds that reflect assumptions made about the additional costs of producing roasted and ground Fairtrade coffee, given the cost of beans and the Fairtrade license. The Fairtrade label thus seems to create a coffee product that roasters and retailers can use to exploit their market power.

**Keywords:** coffee supply chain; fair trade; Fairtrade; market power

**JEL Classification:** D43; O19; P46

## 1. Introduction

Fairtrade certification is a market-based policy instrument aimed to reduce poverty (Fairtrade International 2019). As sales of Fairtrade-certified goods are increasing rapidly (Fairtrade International 2017a), a key question is whether Fairtrade labeling[1] improves market efficiency and welfare or if charity works better, as argued by Weber (2007), Griffiths (2014), Claar and Haight (2015), and De Janvry et al. (2015), among others.

As evident from sales and papers that analyze consumers' willingness to pay a premium for sustainability-certified products, there is overwhelming evidence that many people are willing to pay considerably more for fair trade-certified products (Hainmueller et al. 2015; Basu et al. 2016). Fair trade certification can therefore be viewed as creating a new product (e.g., coffee combined with (perceived) decent incomes and working conditions for poor farmers), that consumers are willing to buy (Reinstein and Song 2012; Dragusanu et al. 2014). Without the certification, the label and the subsequent monitoring (and a presumed positive effect on producers), the market for fair trade products would not exist.

---

[1] The term Fairtrade refers to products certified by Fairtrade International. I use the term fair trade when referring to fair trade programmes in general.

Apart from connecting farmers to consumers, fair trade programs can increase market efficiency by removing other constraints, as there is imperfect competition along most international commodity supply chains (Sexton et al. 2007; Chambolle and Poret 2013). Reinstein and Song (2012) developed a theoretical model for the coffee market showing that fair trade programs can increase market efficiency by inducing farmers' investment in their production process, which reduces the cost for the roaster. Along similar lines, Podhorsky (2015) developed a model of the international coffee supply chain and used it to analyze the impact of reduced market power among oligopolistic intermediaries (the large commodity trading companies). Both studies show that fair trade is more efficient than direct donations if the costs, consisting of farmers' certification costs, processing costs, and the markup on the final goods not received by the farmers, do not exceed the benefits accrued to the farmers. According to Podhorsky (2015), direct transfers would have been more efficient during the global coffee crises 2000–2005, but this is unlikely to be a general feature of fair trade.

The results of Reinstein and Song (2012) and Podhorsky (2015) partly depend on assumptions made about the final goods market, that is, the degree of market power of the roasters and retailers. For example, Podhorsky (2015) assumes there is monopolistic competition and that the price is determined by marginal cost plus a constant markup, which in percentage terms is assumed to be the same for conventional and fair trade coffee. However, markups might differ between fair trade and conventional coffee due to differences in the shape of the demand curve. Furthermore, in most developed countries the consumer markets for coffee are characterized by a few dominant multinational or national roasters, with a combined market share often exceeding 60% (Sutton 2007). Moreover, the grocery retail sectors are usually very concentrated (Bukeviciute et al. 2009; McCorriston 2013). Thus, the price setting of both the roasters and the grocery retail chains, and their relative bargaining power, may impact the markups.

The purpose of the present paper is to evaluate the certification carried out by Fairtrade International by estimating price-cost margins for Fairtrade coffee in the Swedish consumer market. The data available allow me to measure the additional price-cost margin on Fairtrade coffee in Swedish Krona (SEK), but not the price-cost margin on conventional coffee. The additional price-cost margin on Fairtrade coffee can then be compared with the monetary benefits accruing to the producers of coffee beans. Large price-cost margins relative to producers' monetary benefits would be a strong argument in favor of direct transfers to farmers. As far as I know, apart from Podhorsky (2015), who assumes that the markup is the same for conventional and Fairtrade coffee, earlier studies have not attempted to estimate price-cost margins due to lack of information about costs (Samoggia and Riedel 2018; Bissinger 2019).

Although there are many Fairtrade-certified products, coffee is of special interest since it is the most important one: it accounts for about 25% of the value of Fairtrade retail sales and involves over 800,000 Fairtrade coffee farmers (Fairtrade International 2017a, 2017b). In Sweden, it accounts for 27% of Fairtrade's sales (Fairtrade Sweden 2017) and has a market share of about 10% (Fairtrade Sweden 2017), which is similar to figures in several other countries (Krier 2008; Elliott 2012).

In most developed countries, both large- and small-scale roasters produce Fairtrade coffee and primarily sell it through grocery chains. This means that Fairtrade and conventional coffees have the same supply chains after the beans have reached the importing countries. Consequently, Fairtrade coffee is an integrated part of the mainstream coffee retail markets (Fairtrade Foundation 2012; Griffiths 2014).

Coffee retail markets also share many other characteristics in developed countries, so the Swedish market should at least be representative of those in northern Europe (Sutton 2007). In Sweden, two national and two multinational roasters have about 85% of the market, while many small roasters compete for the rest (Durevall 2007; Expertvalet 2018), and the three largest grocery chains account for well over 90% of all sales (Swedish Competition Authority 2011).

The analysis was carried out with scanner data for the period March 2009 to February 2012 at the barcode (EAN) level for all roasted and ground coffee products, collected by the Nielsen company from

over 3000 Swedish grocery shops. First, I estimated the nationally representative average Fairtrade and conventional coffee retail prices by regressing retail prices on a range of product characteristics to control for differences across coffees. Then, to obtain estimates of the Fairtrade premium and its components, I distinguish between three groups of actors: producer countries (i.e., coffee farmers, cooperatives, middlemen, exporters and Fairtrade International); roasters and retailers (in this paper comprising importers, roasters, and retailers in Sweden); and Fairtrade Sweden, which manages the certification of roasters and other related activities in Sweden. I calculated the distribution of shares of the grocery retail-market Fairtrade premium among them using the estimates of average retail prices, the difference in price-cost margins on sales of Fairtrade and conventional coffee, costs of Fairtrade and conventional beans, and Fairtrade license costs, while assuming rather extreme maximum and minimum additional costs related to the production of Fairtrade coffee compared with conventional coffee. The analysis provides an estimate of the size of the price-cost margin on Fairtrade coffee using conventional coffee as the benchmark, so there is no need for information about the costs of producing roasted coffee in general, except for the costs of beans.

The main finding is based on roasted and ground non-organic conventional and Fairtrade coffees. Roasters' and retailers' price-cost margin for Fairtrade coffee ranges from SEK 6.64 (USD 0.95) to SEK 11.10 (USD 1.60) per kg, while the return to producer countries is SEK 3.70 (USD 0.50) per kg. When measured in percentage terms, out of the premium paid for Fairtrade coffee, the net of value-added tax (VAT), and additional costs, roasters and retailers get 61% when additional costs are assumed to be SEK 5/kg (USD 0.70) higher for Fairtrade coffee than for conventional coffee, given the cost of Fairtrade beans and the Fairtrade license, and 70% when the costs are assumed to be the same. Producer countries get 31% or 24%, respectively, and Fairtrade Sweden gets 8% or 6%. An SEK 5/kg (USD 0.70) difference in cost is large and therefore generates a very low lower bound of the share going to roasters and retailers. After all, there are no obvious differences in the marginal cost of production between Fairtrade and conventional roasted and ground coffee. Thus, the Fairtrade program seems to be much less efficient than direct donations. The key reason is the high price-cost margins; assuming a high additional cost of producing Fairtrade coffee reduces them somewhat, but does not make Fairtrade efficient. There are arguably other benefits of Fairtrade certification than monetary rewards (Dragusanu et al. 2014), but the high consumer prices indicate a key weakness of the Fairtrade system.

The next section briefly reviews earlier research on Fairtrade retail prices. Sections 3 and 4 describe the data and method, respectively. Section 5 reports the regression results and calculates the allocations of the Fairtrade returns. Section 6 discusses the findings and Section 7 concludes the paper.

## 2. Earlier Research

The overall goal of Fairtrade is to ensure that producers get a fair price and terms of trade that allow them to improve their lives (Fairtrade International 2019). As a result, a number of studies have evaluated whether farmers benefit from Fairtrade, with varying results.[2] There are also several studies on willingness to pay for fair trade-labeled products. Although the studies differ in various ways, they clearly show that consumers are prepared to pay more for sustainability-certified products (Basu and Hicks 2008; Hertel et al. 2009; Carlsson et al. 2010; Howard and Allen 2010; Hiscox et al. 2011; Andorfer and Liebe 2012; Van Loo et al. 2015; Hainmueller et al. 2015; Basu et al. 2016). Some of these studies use real-life experiments. One example is Hiscox et al. (2011), who, by setting up eBay

---

[2]   Nelson and Pound (2009) conclude that Fairtrade producers enjoy higher returns and more stable incomes than others. Dragusanu et al. (2014) agree but note that the empirical evidence is based primarily on conditional correlations, while Mohan (2010), Blackman and Rivera (2011), and Dammert and Mohan (2015) argue that there is a lack of persuasive evidence that coffee certification provides significant economic benefits. Recent empirical studies on the impact of Fairtrade on coffee farmers are Weber (2011); Jena et al. (2012); Dragusanu and Nunn (2018); Chiputwa et al. (2015); De Janvry et al. (2015); Minten et al. (2015); and Nelson et al. (2016). Their findings are mixed, yet none of them reports strong positive average effects on income and other indicators of standard of living. In a study commissioned by Fairtrade, Darko et al. (2017) review recently published research and conclude that contextual factors affect the impact on producer markets.

auctions with products that are identical except for a fair trade label, showed that US consumers are willing to pay approximately 23% more for fair trade coffee.

There is also a small number of studies that aim to quantify the average size of the premium by estimating hedonic models. Three recent ones are Bosbach and Maietta (2019), who found that Fairtrade labels increase the price by about 30% in the Italian market, Bissinger and Leufkens (2017) who estimated the premium to 55% in Germany, and Wang (2016), who found that the fair trade labels increased prices by 15–30% in the US market. There is one study on Sweden (Schollenberg 2012), which used Nielsen data for March 2005–March 2008. A Fairtrade label raised the price by 32% when controlling for a range of factors that influence prices, including brands. A drawback of the study is that by controlling for practically all brands, several of which have Fairtrade coffees, it is not clear what Fairtrade products the 32% applies to.

There are also studies that focus on the difference between Fairtrade consumer and bean prices. To the best of my knowledge, only five published papers actually measured the premium paid or provide some information about how it is distributed (Mendoza and Bastiaensen 2003; Kilian et al. 2006; Johannessen and Wilhite 2010; Valkila et al. 2010; Hejkrlík et al. 2013). Two of them, Mendoza and Bastiaensen (2003) and Kilian et al. (2006), analyzed data from the 1990s and early 2000s, when Fairtrade coffee was mostly sold in specialty shops and its market share was tiny (Fairtrade Foundation 2012). Since there has been a rapid increase in the sales of Fairtrade coffee and a shift to grocery chains, these studies are probably no longer relevant (Smith 2009; Mohan 2010, pp. 52–55).

Johannessen and Wilhite (2010) analyzed 2006–2007 data for a Fairtrade coffee sold in retail stores in Norway: Farmers' Coffee from Guatemala. They found that out of the final consumer price, the retailer received 13.8%, the Fairtrade certifier 2.4%, and the importer/roaster 58.2%. This implies that 74.4% of the value stays in Norway while 26.6% ends up in Guatemala. There is no information about production costs, VAT, etc., and no comparison with prices of conventional coffee, so we cannot say anything about the premium paid for Fairtrade coffee or how it was distributed.

Valkila et al. (2010) compared the prices in 2006–2009 of the two most popular Fairtrade coffees with those of the four most popular conventional coffees sold by a large retail chain in Finland. They found that 35% of the Fairtrade consumer price goes to the bean producer country, while 60% stays in Finland. The remaining 5% cover license fees and transport costs. The producer country received EUR 1.30/kg of Fairtrade coffee and EUR 1.15/kg of conventional coffee, which implies that 11.5% of the premium paid by consumers for Fairtrade coffee reaches the producer country. However, the conventional coffees are likely to have low prices, since they are the most popular ones, so the benchmark used for conventional coffee might be biased downwards.

Hejkrlík et al. (2013) analyzed data from the Czech Republic for a large number of ground coffee products. They found that consumers pay 32% more for Fairtrade coffee than conventional coffee, out of which Fairtrade farmers' social premium is 2 percentage points. However, they did not distinguish between ground coffee sold in coffee shops and fair trade shops, where prices presumably are high, from coffee sold in grocery stores, and they did not distinguish between organic and non-organic Fairtrade coffees in their regressions. Moreover, they did not use data on the cost of production.

Thus, no study analyzes a representative sample from a market where large companies produce Fairtrade coffees and grocery chains sell them, and no study attempts to estimate markups on Fairtrade coffee.

## 3. Data Description

The data on coffee products are from weekly sales in 3088 Swedish grocery shops from 1 March 2009 to 26 February 2012, collected at the barcode level by the Nielsen company. They include values and volumes of all coffee products sold as well as information about types of coffee and various product characteristics, such as manufacturer, type of roast, size of package, organic, Fairtrade, and private label (retailer-owned brand). I measured retail prices as value divided by volume averaged over the

sample period and grocery shops (Nielsen does not provide information from the individual grocery shops in Sweden). I focused on roasted and ground coffee, as this is by far the largest market segment, accounting for 80% of all coffee sales in value terms according to the Nielsen data. Instant coffee, which accounts for 11% of sales, is more challenging to analyze due to the small number of Fairtrade products and larger scope for using cheap beans.[3]

Table 1 provides price information on the 188 ground coffee products available in packages of 250 g, 400–499 g, and 500 g. There are 22 organic Fairtrade and 12 organic non-Fairtrade products. To be certified as organic, the coffee has to be grown without chemical pesticides or fertilizers and be untreated with preservatives and other chemicals. For Fairtrade certification, producers have to comply with Fairtrade Standards, which aim to ensure the economic, social, and environmental development of the producers' families, communities, and organizations. The Fairtrade Standards include some environmental criteria, but the key feature is minimum prices and the premium paid on top of world markets prices. An additional premium is paid for organically certified coffee (Fairtrade Foundation 2012).

**Table 1.** Ground coffee and bean prices, March 2009–February 2012 (USD in parentheses).

| Ground Coffee | Number | Mean | Median | Min | Max |
|---|---|---|---|---|---|
| Conventional | 151 | 71.20 (10.20) | 64.08 (9.20) | 30.00 (4.30) | 175.93 (25.10) |
| Fairtrade organic | 22 | 107.16 (15.30) | 90.80 (13.00) | 69.32 (9.90) | 185.71 (26.50) |
| Fairtrade not organic | 3 | 121.00 (17.30) | 95.00 (13.60) | 82.00 (11.70) | 186.00 (26.60) |
| Organic, not Fairtrade | 12 | 71.69 (10.20) | 67.54 (9.70) | 41.02 (5.90) | 129.44 (18.50) |
| Green beans | | Ordinary | Fairtrade | | |
| Import price per kg Import price of 1 kg of green beans | | 29.72 (4.30) | 32.80 (4.70) | | |
| Cost of producing 1 kg roasted coffee | | 35.37 (5.00) | 39.03 (5.60) | | |

Note: Based on data from Nielsen and Statistics Sweden. The cost of producing 1 kg roasted coffee is higher than the price of 1 kg of beans because of weight lost during roasting. Approximately 1.19 kg of green beans are used to produce 1 kg of ground coffee (European Coffee Federation 2014).

Both the mean and median prices of organic Fairtrade coffee are relatively high, 30–40% higher than for conventional coffee. This is partly due to the low prices of conventional coffee at the lower end of the price scale; the minimum price is SEK 30 (USD 4.30) compared with SEK 69 (USD 9.90) for Fairtrade coffee while the maximum price for conventional coffee is only SEK 10 (USD 1.40) lower than for Fairtrade coffee. The price difference is probably due to the Fairtrade label since organic non-Fairtrade coffee is only slightly more expensive than conventional coffee. There are three non-organic Fairtrade coffee products. They are in the 250 g segment, which is very small; it only accounts for 0.5% of total sales of ground and roasted coffee. The prices of the three non-organic Fairtrade coffees also differ greatly and are therefore not useful for estimating representative Fairtrade coffee prices (see Appendix A).

There are two sources of information about green bean prices: International Coffee Organization (ICO) and Statistics Sweden. ICO publishes daily world market prices for various types of green coffee beans. I used these prices and information on the volume of imports of green beans to construct an index with weights based on the type of Arabica beans imported (European Coffee Federation 2014). Statistics Sweden publishes monthly volumes and values of imports of green beans. The average bean prices obtained from the two sources are very similar, SEK 29.15 and 29.72/kg for March 2009–February 2012. The difference is probably due to quality differences and additional freight and insurance costs for delivery to Sweden. Converting the freight and insurance costs used by Valkila et al. (2010) from

---

[3] I analyzed instant coffee and the results are available on request. They are qualitatively similar to the ones reported for ground coffee.

EUR to SEK gives a cost of SEK 0.85/kg (USD 0.12) for transporting green beans from Latin America to Finland, so the SEK 0.57 (USD 0.08) difference between Statistics Sweden and ICO prices makes sense; ICO prices are for delivery to the US, France, or Germany. Since the difference in the prices is small, the choice of data source does not matter for the results. In the calculations, I used prices based on import data from Statistics Sweden.

The current Fairtrade (minimum) bean price is 140 US cents per pound for washed Arabica, 135 US cents per pound for natural Arabica, and 101 US cents per pound for Robusta. On top of those prices, Fairtrade requires buyers to add a social premium of 20 US cents per pound to the price of conventional coffee beans and another 30 US cents for certified organic coffee beans (Fairtrade Foundation 2012).[4] Since world market prices were higher than Fairtrade minimum prices during the study period, I added the Fairtrade social premiums to world market prices to obtain the prices paid for Fairtrade coffee beans. Most Fairtrade coffee sold in Sweden is organic, so Table 1 reports both Fairtrade and organic Fairtrade bean prices. Unfortunately, I do not have systematic information about organic non-Fairtrade bean prices.

The price of ordinary green beans was SEK 29.72/kg (USD 4.25) during the study period. Adding Fairtrade's social premium increases this figure to SEK 32.80/kg (USD 4.70) for Fairtrade beans and SEK 37.50/kg (USD 5.50) for organic Fairtrade beans. This means that Fairtrade and organic Fairtrade beans are 10% and 26% more expensive than ordinary beans, respectively.

Approximately 1.19 kg of green beans is used to produce 1 kg of ground coffee due to weight lost during roasting (European Coffee Federation 2014). When comparing green bean and ground coffee prices, it therefore makes sense to multiply bean prices by 1.19. Roasters thus paid SEK 35.37 (USD 5.10) for the beans used to produce 1 kg of ground conventional coffee. For Fairtrade coffee and organic Fairtrade coffee, the corresponding figures were SEK 39.03 (USD 5.60) and SEK 44.63 (USD 6.40).

We can conclude that there are large retail price differences between conventional and Fairtrade coffee and that these are unlikely due to differences in bean prices only. However, the comparisons ignore the fact that the coffees compared are not identical; many characteristics of the products affect price, such as size of packages and type of roasting.

## 4. Method

In this section I explain how the price-cost margins are estimated. In the first step, regression analysis is used to obtain average prices for Fairtrade and conventional coffee, while controlling for various indicators of quality, and in the second step, data on prices and costs are used to calculate how the premium paid for Fairtrade in the Swedish market is distributed.

One challenge is that the quality might differ both between conventional and Fairtrade coffees and within each category (Elliott 2012). Another is that almost all Swedish Fairtrade coffees are organic, and the respective contributions of Fairtrade and organic beans to the price need to be disentangled; there is a paucity of prices of organic beans imported to Sweden, precluding an analysis of organic coffees. With regression analysis, I can estimate the average price paid for Fairtrade and conventional coffee, while controlling for several product characteristics. It shows how much the Fairtrade label and the organic labels add to the price of a standard package of branded coffee, given a number of quality indicators.

To identify the impact of a Fairtrade label on the price, I used the fact that not all organic coffees are Fairtrade and that the price of organically certified conventional coffee should be informative about the contribution of organic coffee labels to the price of organically certified Fairtrade coffee. The main analysis is restricted to coffees sold in 500 g packages. This is by far the most popular package size, making up 93% of all sales of roasted coffee in value terms during the study period, according

---

[4] Before April 2011, the social premiums were 10 and 20 US cents, respectively.

to the Nielsen data. In this market segment, all Fairtrade coffees are organic. However, to check for robustness of the results, I also estimated models with various sub-samples (reported in Appendix A).

The second step involves calculating the values of the price-cost margins of roasters/retailers for Fairtrade coffee and comparing them with the returns obtained by producer countries. I first used a simplified model to show the basic approach and then provide a more detailed description.

Following Podhorsky (2015), we assume there is imperfect competition in the Swedish coffee market (i.e., that price is equal to marginal cost multiplied by a price markup). Retail prices and some of the marginal cost components, such as the Fairtrade certification fee, $P_{FTC}$, and the cost of Fairtrade beans, $P_{BF}$, and conventional beans, $P_{BC}$, are known, while most other costs, such as wage, packaging and transport costs, $P_O$, can be assumed to be the same for Fairtrade and conventional coffee. There might also be a cost of production of Fairtrade coffee, $P_{AF}$, in addition to the cost of Fairtrade beans and the Fairtrade fee. This is unknown, so assumptions about its minimum and maximum values are used in the analysis.

The retail price in SEK per kg of Fairtrade coffee, $P_{RF}$, and conventional coffee, $P_{RC}$, are given by

$$P_{RF} = (1 + MU + MU_F)(P_{BF} + P_{FTC} + P_{AF} + P_O) \tag{1}$$

$$P_{RC} = (1 + MU)(P_{BC} + P_O), \tag{2}$$

where $MU$ is the markup on roasted coffee in general and $MU_F$ a markup that potentially is charged on Fairtrade coffee in addition to $MU$. Reinstein and Song (2012) and Podhorsky (2015) assumed that $MU_F$ is zero or very small, implying that the premium paid for Fairtrade coffee by consumers only reflects the extra production costs.

By rearranging terms in Equations (1) and (2), we get

$$P_{RF} = P_{BF} + P_{FTC} + P_{AF} + P_O + MU(P_{BF} + P_{FTC} + P_{AF} + P_O)$$
$$+ MU_F(P_{BF} + P_{FTC} + P_{AF} + P_O) = P_{BF} + P_{FTC} + P_{AF} + P_O + MA_{GF} + MA_F \tag{3}$$

$$P_{RC} = P_{BC} + P_O + MU(P_{BC} + P_O) = P_{BC} + P_O + MA_G, \tag{4}$$

where $MA_G$ and $MA_{GF}$ are price-cost margins in SEK per kg due to the general markup and Fairtrade markup.

The difference between $P_{RF}$ and $P_{RC}$ is

$$P_{RF} - P_{RC} = P_{BF} - P_{BC} + P_{FTC} + P_{AF} + MA_{GF} - MA_G + MA_F \tag{5}$$

where the unobserved other costs, $P_O$, cancel out. Equation (5) shows that apart from the components of the price-cost margins, only the additional cost of producing Fairtrade coffee, $P_{AF}$, is unknown.

The aim of the analysis is to estimate $(MA_{GF} - MA_G) + MA_F$, which I call the excess price-cost margin on Fairtrade coffee. It consists of the difference in SEK resulting from the higher cost of production of Fairtrade beans plus license fee and the use of a general (constant) markup applied to all coffees, $MA_{GF} - MA_G$, and the additional margin due to the (potentially) additional markup on Fairtrade coffee, $M_F$. Given my data, it is a challenge to disentangle the two components; only minimum and maximum values of $(MA_{GF} - MA_G) + MA_F$ can be estimated. However, these values are of key interest since they indicate how much of the consumer market Fairtrade premium ends up in the pockets of roasters and retail chains. Fairtrade increases market efficiency if $(MA_{GF} - MA_G) + MA_F$, plus other costs associated with Fairtrade, are smaller than the benefits accruing to the Fairtrade farmers (Reinstein and Song 2012; Podhorsky 2015).

It is straightforward, in principle, to obtain values for $(MA_{GF} - MA_G) + MA_F$, but it involves a number of steps. The two main issues are how to treat VAT, which has been ignored so far and is the government's income in the form of tax, and what to assume about potential differences in production costs between Fairtrade and conventional coffee in addition to the cost of beans and license fees, $P_{AF}$.

In the calculations described below, all prices and costs are measured inclusive of VAT. In Sweden, the VAT on food is 12% (i.e., 10.7% of the price inclusive of VAT). The rate for other products relevant for the study is 25%, which equals 20% of the price including VAT. All prices and costs are in SEK per kg of coffee. USD 1 equaled about SEK 7 during 2009–2012.

The maximum values of $P_{AF}$ are based on an analysis of wholesale prices and the cost of imported beans. I used SEK 5/kg (USD 0.70) as the maximum difference in additional costs between Fairtrade and conventional coffee, which is likely to be much higher than the actual difference. The minimum value is zero, since the production processes for Fairtrade and conventional roasted coffee are for all practical purposes the same. To obtain the maximum value, I used annual data from Statistics Sweden on values and volumes of deliveries from Swedish roasters and the price of imported green coffee beans. The average wholesale delivery price in 2010–2011 was SEK 52/kg (USD 7.40), while the import price of coffee beans was SEK 32/kg (USD 4.60). The difference, SEK 20 (USD 2.85), consists of the margin plus costs for roasting (including bean weight loss, which corresponds to SEK 6 (0.85)), packaging, transporting to retailers, etc. A difference in production costs between Fairtrade and conventional coffee of SEK 5/kg (USD 0.70) should therefore be a very high upper limit.

The following steps explain the calculations:

Roasters'/retailers' price-cost margin, $MA_i$, on 1 kg of roasted and ground conventional or Fairtrade coffee is

$$MA_i = P_{Ri} - (0.107P_{Ri} - 0.107P_{Bi} - 0.107P_{FTC} - 0.2P_{Oi} - 0.2P_{AF}) - P_{Bi} - P_{FTC} - P_{Oi} - P_{AF},$$

where $i$ is $C$ for conventional coffee and $F$ for Fairtrade coffee. The term in parentheses is the net payment of VAT.

–  The import price for beans inclusive of VAT and weight lost is $P_{Bi} = 1.19P_{impi}$, where $P_{impi}$ is the border price of conventional or Fairtrade green beans and 1.19 is the adjustment due to weight lost ([European Coffee Federation 2014](#)).

–  Producer countries' income from sales of Fairtrade beans is $0.893(P_{BF} - P_{BC})$, where 0.893 removes the VAT from the difference in prices of Fairtrade and conventional beans.

–  Fairtrade Sweden's income from the certification fee is $P_{FTC} = 0.893(0.015 * 0.893P_{RF})$. Fairtrade Sweden gets the net of VAT value of SEK 0.015 times the net of VAT retail price.

–  Roasters'/retailers' excess margin on Fairtrade coffee, $(MA_{GF} - MA_G) + MA_F$, is $MA_F - MA_C$, i.e., the difference between the margins from sales of Fairtrade and conventional coffee:

$$MA_F - MA_C = (P_{RF} - P_{RC}) + (P_{BC} - P_{BF}) + 0.107(P_{RC} - P_{RF}) + 0.107(P_{BC} - P_{BF})$$
$$-0.893P_{FTC} - 0.893P_{AF}$$

–  To calculate how the Fairtrade premium in the consumer market is distributed, a measure of total income from Fairtrade retail sales is needed. It is defined as $R_{FTR} = P_{FTC} + 0.893(P_{BF} - P_{BC}) + (MA_F - MA_C)$. Thus, income from Fairtrade is made up of the license fee plus the income of the producer country and roasters'/retailers' excess margin from sales of Fairtrade coffee. The government's income is ignored since the VAT is small.

–  Roasters'/retailers' share of the total additional value of Fairtrade retail sales is $(MA_F - MA_C)/R_{FTR}$.

–  Producer countries' share of the additional income from Fairtrade retail sales is $\left[0.893(P_{BF} - P_{BC})\right]/R_{FTR}$.

–  Fairtrade Sweden's share of the additional income from Fairtrade retail sales is $P_{FTC}/R_{FTR}$.

## 5. Results

*5.1. How High Are Fairtrade Prices?*

Table 2 reports Ordinary Least Squares (OLS) = regressions on prices per kg of ground coffee in 500 g packages, using robust (sandwich estimator) standard errors. Product characteristics, aimed at capturing quality-related costs, are measured by dummy variables for type of roast (medium, dark, and other), private label, decaffeinated, organic (not Fairtrade), and Fairtrade organic coffee (all 500 g Fairtrade coffees are organic). The dummies are not mutually exclusive; a small number of coffees with private labels are also organic and a few are both organic and Fairtrade. However, the inclusion of more dummy variables, such as non-Fairtrade-organic private label, does not affect the results (available from the author on request).

**Table 2.** OLS regression on average price per kg of ground coffee (500 g packages).

|  | (1)<br>All Products | (2)<br>No Fairtrade | (3)<br>Only Organic |
|---|---|---|---|
| Dark roast | 5.05 | 4.78 | 7.54 |
|  | (1.97) * | (1.68) * | (3.99) *** |
| Undefined roast | 17.45 | 17.333 | −5.07 |
|  | (1.85) * | (1.84) * | (1.62) |
| Decaffeinated | 6.40 | 6.31 |  |
|  | (0.31) | (0.30) |  |
| Private label | −11.55 | −11.70 | −9.72 |
|  | (4.96) *** | (4.76) *** | (2.61) ** |
| Fairtrade organic | 23.27 |  | 14.76 |
|  | (11.90) *** |  | (5.25) *** |
| Organic, not Fairtrade | 6.14 | 6.17 |  |
|  | (1.74) * | (1.75) * |  |
| Constant | 62.00 | 62.14 | 69.26 |
|  | (26.03) *** | (25.18) *** | (22.09) *** |
| $R^2$ | 0.29 | 0.20 | 0.80 |
| $N$ | 140 | 127 | 24 |

Note: Average price for 1 March 2009–26 February 2012. Robust standard errors are used. * $p < 0.1$; ** $p < 0.05$; *** $p < 0.01$. USD 1 = Swedish Krona (SEK) 7.

Specification 1 includes the 140 products for which there is data. The base category is ground medium-roast branded coffee with caffeine. The price of it is SEK 62.00/kg (USD 8.90). The combined Fairtrade and organic labels add SEK 23.27/kg (USD 3.30) to the SEK 62.00/kg (USD 8.90), while organic coffee labels by themselves add only SEK 6.14/kg (USD 0.9). The estimate of the contribution of organic beans to the price is somewhat uncertain because it is only significant at the 10% level, but it is clearly much smaller than the SEK 17/kg (USD 2.40) (23.27 minus 6.14) contribution of the Fairtrade label. It is noteworthy that Bosbach and Maietta (2019) report a similar price difference using Italian data.

Since the variable measuring Fairtrade coffee products includes only organic Fairtrade coffee, I re-estimated the model without Fairtrade coffee to focus on organic coffee (specification 2). The results are similar: organic beans add SEK 6.17/kg (USD 0.90) to the price. To check the robustness of the result for the Fairtrade coffee, I then estimated a model with only organic coffee (specification 3). Now the base category is a 500 g package of ground medium-roast organic branded coffee with caffeine, priced at SEK 69.26/kg (9.90). There are only 24 observations, but the results are strong: the coefficient for Fairtrade coffee is highly significant (t-value = 5.25), showing that the Fairtrade label adds SEK 14.76/kg (USD 2.10) to the price of organic coffee. This is in line with the results obtained in the two other specifications. A medium-roast branded coffee with caffeine that is also Fairtrade but not organic would thus cost about SEK 77–79 (USD 11.00–11.30). This implies that the Fairtrade label increases the price of conventional coffee by about 25%.

All the control variables have expected signs. Private label coffee is about SEK 12 (USD 1.70) cheaper than branded coffee, and dark roast is SEK 5–7 (USD 0.70–1.00) more expensive. The 'undefined roast' is a control variable that captures products without a type of roast identified on the package. Decaffeinated coffee is SEK 6 (USD 0.85) more expensive than conventional coffee, but the estimates are not significant due to the small number of observations.

To further check for robustness of the findings, I estimated models with five different samples: all 250 g, 400–499 g, and 500 g packages; only 400–499 g and 500 g packages, which exclude several very expensive 250 g packages; the four roasters that dominate the Swedish market for ground coffee; only inexpensive coffees, i.e., coffees that cost less than SEK 100/kg (USD 14.25); and only branded coffees. Although the coefficients of some of the product characteristics differ, the ones for organic and Fairtrade coffee are similar to the coefficients reported in Table 2 (see Table A1 in Appendix A).

*5.2. Distribution of the Premium Paid for Fairtrade Coffee*

The purpose of this section is to calculate how much of the premium consumers pay for Fairtrade coffee that accrues to roasters and retailers, Fairtrade Sweden, and producer countries.

During the study period, the average Fairtrade social premium (i.e., the additional cost roasters pay for green Fairtrade beans), was SEK 3.11/kg (USD 0.45) on ordinary beans (see Table A2 in Appendix A). The certification fee paid by roasters was 1.5% of the consumer price in 2008 and 0.8% in 2013.[5] In the calculations, I used 1.5% (inclusive of VAT) of the consumer price (exclusive of VAT), which might be on the high side.

As there is a time lag between the purchase of beans and the sale of processed coffee, I calculated average bean prices starting three and six months before the study period, as well as in March 2009, the first month of the sample. However, the price changes were small, and the choice did not matter much. The price used in the calculations was the average price of imported green beans for January 2009–November 2011.

Table 3 reports the results for ground coffee in 500 g packages. Consumer prices are from Table 2, specification 1, where I controlled for product characteristics. They were SEK 62 (USD 8.90) for conventional coffee and SEK 79 (USD 11.30) for Fairtrade coffee. Out of the premium paid for 1 kg of Fairtrade coffee, producer countries and Fairtrade Sweden receive SEK 3.70/kg (USD 0.50) and 0.95/kg (USD 0.95), respectively. When I assumed that 'additional costs' are zero, the roasters' and retailers' margin was SEK 11.10/kg (USD 1.60) higher for Fairtrade than conventional coffee. The total additional value of Fairtrade sales was thus SEK 15.75/kg (USD 2.25), of which 24% accrued to producer countries, 70% to roasters and retailers, and 6% to Fairtrade Sweden. If we instead assumed that 'additional costs' are SEK 5/kg (0.70) higher for Fairtrade than conventional coffee, the roasters' and retailers' excess margin was SEK 6.64/kg (USD 0.95). The share for producer countries increased to 31% and the share for roasters and retailers decreased to 61%.

Table 3 also reports how large the difference in 'additional costs' needs to be to completely erode roasters' and retailers' excess margin; it is SEK 14.80/kg (USD 2.10). In this hypothetical case, roasters' and retailers' margins in SEK are the same for conventional and Fairtrade coffee, and producer countries' share is 80%. If the markup is constant, a part of the SEK 14.80/kg (USD 2.10) is due to higher costs, which could be viewed as payment for fixed costs associated with joining Fairtrade. Still, SEK 14.80/kg (USD 2.10) is a large sum compared with SEK 3.70/kg (USD 0.50), the premium paid on Fairtrade beans.

---

[5]    Personal communication with Morgan Zerne, CEO of Fairtrade Sweden.

**Table 3.** Roasters'/retailers' excess margin and distribution of premium paid for Fairtrade roasted and ground coffee, in SEK/kg (500 g packages).

| Measure | Definition | Conventional Coffee | Fairtrade, Other Costs = SEK 0 | Fairtrade, Other Costs = SEK 5 | Fairtrade, Other Costs = SEK 14.80 |
|---|---|---|---|---|---|
| Retail price | $P_{Ri}$ | 62.00 (8.90) | 79.12 (11.30) | 79.12 (11.30) | 79.12 (11.30) |
| Cost of beans (inclusive of value-added tax (VAT) and weight lost) | $P_{Bi} = P_{impi}(1.12)(1.19)$ | 37.95 (5.40) | 42.09 (6.00) | 42.09 (6.00) | 42.09 (6.00) |
| Fairtrade additional cost (assumed) | $P_{Ai}$ | 0.00 | 0.00 | 5.00 (0.71) | 14.80 (2.10) |
| Roasters'/retailers' excess margin | $MA_F - MA_C$ | - | 11.10 (1.60) | 6.64 (0.95) | 0.00 |
| Producer countries' income | $0.893(P_{BF} - P_{BC})$ | - | 3.70 (0.50) | 3.70 (0.50) | 3.70 (0.50) |
| Fairtrade Sweden's income | $P_{FTC} = 0.893(0.015(P_{RF}/1.12))$ | - | 0.95 (0.15) | 0.95 (0.15) | 0.95 (0.15) |
| Total additional income from Fairtrade sales | $R_{FTR}$ | - | 15.75 (2.25) | 11.29 (1.60) | 4.65 (0.65) |
| Producer countries' share | $\left[0.893(P_{BF} - P_{BC})\right]/R_{FTR}$ | - | 24% | 31% | 80% |
| Fairtrade Sweden's share | $(0.893P_{FTC})/R_{FTR}$ | - | 6% | 8% | 20% |
| Roasters'/retailers' share | $[(M_F - M_C)]/[R_{FT} + (P_{NF} - P_{NC})]$ | - | 70% | 61% | 0% |
| Sum of shares | | - | 100% | 100% | 100% |

Note: The price data are from Table 2. Based on average retail prices for 1 March 2009–26 February 2012, and average prices of imported green beans for January 2009–November 2011.

## 6. Discussion

By comparing conventional and Fairtrade coffee and assuming that it costs SEK 5/kg (USD 0.70) more to produce Fairtrade than conventional coffee (which is a very high value), I found that the Fairtrade coffee price-cost margin for roasters and retailers was SEK 6.64/kg (USD 0.95) and the return to producer countries was SEK 3.70/kg (USD 0.55). This implies that roasters and retailers get 61% of the premium, while producer countries get 31%. Assuming that the cost of producing Fairtrade and conventional coffee is the same, apart from the price of beans and licenses, changes the distribution to 70% and 24%, respectively. The share going to Fairtrade Sweden is 8% in the first case and 6% in the second case. These percentages can be considered lower and upper bounds. Thus, during the period studied, the Fairtrade premium received by the farmers was clearly lower than the price-cost margin in the Swedish market. The Fairtrade label therefore seems to create a product that roasters and retailers can use to exploit their market power.

Fairtrade critics suggested that consumers should donate money to coffee farmers (supposedly via some institution) instead of buying Fairtrade coffee (Weber 2007; Griffiths 2014; Leclair 2002; De Janvry et al. 2015; Claar and Haight 2015). Reinstein and Song (2012) and Podhorsky (2015) developed models that show the conditions required for Fairtrade to increase market efficiency. They are clearly far from fulfilled in the Swedish coffee market, as roasters' and retailers' excess margins for Fairtrade are large relative to the producer countries' (and thus farmers') benefits. Although there are additional advantages of being a Fairtrade farmer besides monetary rewards, these are likely to be small (De Janvry et al. 2015; Dragusanu and Nunn 2018) compared with the total cost of Fairtrade, which includes certification fees in producer and consumer countries, additional processing costs, as well as the excess margins. Thus, there is evidence that the benefits of direct transfers in the form of charity might be larger than those provided by Fairtrade.

One suggestion is that roasters should offer two coffees that are identical in all respects except that one has a sum printed on it, which is donated to charity projects in poor countries (Griffiths 2010). However, this would require monitoring by an external agency to be credible and would remove the perceived link between consumption and coffee production with decent pay and working conditions. Moreover, the proposal to use donations is unlikely to have much policy relevance, since Fairtrade is a well-known organization with a well-established label. In addition, consumers seem to be more willing to donate indirectly by paying a price premium on Fairtrade coffee than buying conventional coffee and giving a charity donation (Koppel and Schulze 2013). A better strategy would be to improve the functioning of the Fairtrade system. One suggestion is that Fairtrade should require roasters to declare the difference between Fairtrade and conventional beans on packages. Although this would only give a rough indication of how much of the retail price goes to Fairtrade cooperatives, as prices vary, it would put a limit on how much roasters can charge for Fairtrade coffee. If unattractive to roasters in general, it could be adopted by those seriously engaged in Fairtrade, increasing their market shares and boosting competition in the coffee market.

One limitation of the study is the focus on the Swedish retail market. Although the Swedish coffee retail market is likely to be representative of most other coffee retail markets in some aspects, such as market structure (Sutton 2007), it differs from those outside northern Europe in several ways. Most importantly, per capita consumption is very high; Arabica beans are much more popular than Robusta beans, and instant coffee has a relatively small market share. Yet, studies that used hedonic models, such as Bosbach and Maietta (2019) and Wang (2016), found that excess payment for fair trade coffee was similar in the Italian and US retail markets to what I found in the Swedish market.

Another limitation is the focus on coffee. It is possible that the excess payments on more homogenous Fairtrade products, such as bananas and flowers, are low compared to coffee; it is easier for consumers to compare price and quality when products are similar. Nevertheless, Bissinger (2019) found that producer prices on Fairtrade-certified cereals, fiber crops, and vegetables are twice as high as prices on similar conventional products. Thus, more research is required to ensure that results are valid for Fairtrade products in general and Fairtrade coffee sold in markets outside of northern Europe.

## 7. Conclusions

The sale of Fairtrade products has grown rapidly in recent years, partly due to increased consumer demand and partly due to certification of Fairtrade cities, towns, and regions, which are committed to the promotion and procurement of Fairtrade certified goods. Since there is a link between the 2030 Agenda for Sustainable Development, launched by the United Nations in 2015, and fair trade, there is reason to expect further increases in both consumer demand and political support for fair trade products.

Therefore, an important question is if Fairtrade certification is efficient, that is, whether producers benefit more from participating in Fairtrade than from donations of the excess payment made on Fairtrade products. This paper analyzes the efficiency of Fairtrade certification of coffee in the Swedish retail market. A key finding was that when we assume that it costs SEK 5/kg (USD 0.70) more to produce Fairtrade than conventional coffee, over and above the cost of beans and the Fairtrade license, roasters and retailers get 61% of the premium, while producer countries get 31%. The share going to Fairtrade Sweden is 8%. A difference of SEK 5/kg is an extreme assumption so 31% can be viewed as a maximum share. Therefore, Fairtrade certification of coffee does not seem to be efficient as most of the excess payment accrues to roasters and/or grocery chains. Although the study is limited to sales of roasted and ground Fairtrade coffee in Sweden, the results are strong and probably relevant for other coffee retail markets.

**Funding:** Financial support from the Swedish Competition Authority is gratefully acknowledged.

**Acknowledgments:** I am grateful for useful comments by Arne Bigsten, Sven-Olof Daunfeldt, Niklas Rudholm, Rohini Somanathan, Måns Söderbom and Debbie Axlid.

**Conflicts of Interest:** The author declares no conflicts of interest.

## Appendix A

*Additional Tables*

Table A1 reports regressions with various samples to show that the findings are not likely to be due to outliers or extrapolation, i.e., comparisons of completely different products. Five samples are used: 250, 400–499 g, and 500 g packages; 400–499 g and 500 g packages; only the four largest roasters; only inexpensive coffee, price < SEK 100/kg; and only coffee with national labels.

The estimated coefficients for Fairtrade and organic coffee are similar to the ones in Table 2. The most notable result is that the four large roasters have a somewhat higher price for the base category (SEK 64.60 vs. SEK 62.00) and add fewer SEK to Fairtrade and organic coffee. However, using these prices only marginally affects the distribution of shares; for instance, roasters/retailers get 57% instead of 61% when we assume that the difference in 'other costs' is SEK 5/kg. Another result is that coffees in 250 g and 400–499 g packages are much more expensive than those in the standard 500 g package. However, they make up a small and heterogeneous group. For example, in the 250 g group there are only 23 products. Prices range from SEK 52 to 186/kg, and the average price is SEK 51 higher per kg than the average price of 500 g packages. The 250 g group also includes three non-organic Fairtrade coffees, which were excluded from the sample because there are too few to provide a reliable estimate, and there are no non-organic Fairtrade coffees in the other categories. The prices of the three 250 g non-organic Fairtrade coffees are SEK 82, 95, and 186/kg.

The estimates of the coefficients for the control variables vary in some cases, particularly those of decaffeinated coffee and undefined roasts. This is primarily due to few observations.

Table A2 reports Fairtrade's social premiums on world market prices of green beans, converted into SEK per kg for the production of 1 kg of ground coffee. As is evident, there was a sharp increase in premiums in March 2011. The values in Table A2 are used to calculate the cost of Fairtrade beans in Table 3.



**Table A1.** OLS regressions on average price of ground coffee, various samples (USD 1 = SEK 7).

| | 250–500 g Packages | 400–500 g Packages | Four Large Roasters only (400–500 g) | Only with Price <SEK 100/kg (400–500 g) | No Private Label |
|---|---|---|---|---|---|
| Dark roast | 8.474 | 6.921 | 5.183 | 6.129 | 6.796 |
| | (3.43) *** | (2.80) *** | (2.06) ** | (2.61) ** | (2.27) ** |
| Undefined roast | 20.408 | 20.478 | −7.766 | 7.344 | 22.440 |
| | (2.69) *** | (2.32) * | (3.91) *** | (1.20) | (2.41) ** |
| Decaffeinated | 7.182 | 1.442 | 0.639 | 7.786 | 1.733 |
| | (0.58) | (0.09) | (0.25) | (0.64) | (0.11) |
| Private label | −14.026 | −10.843 | | −10.957 | |
| | (5.62) *** | (4.62) *** | | (5.06) *** | |
| 250 g | 51.594 | | | | |
| | (10.18) *** | | | | |
| 400–499 g | 22.734 | 24.277 | 12.792 | 15.962 | 24.369 |
| | (5.36) *** | (5.63) *** | (3.90) *** | (4.83) *** | (5.59) *** |
| Fairtrade organic | 27.629 | 26.780 | 21.631 | 23.412 | 27.241 |
| | (7.92) *** | (7.52) *** | (9.19) *** | (12.17) *** | (6.60) *** |
| Organic only | 7.342 | 6.907 | 5.935 | 7.025 | 4.373 |
| | (2.00) ** | (1.96) * | (2.05) ** | (2.23) ** | (0.82) |
| Constant | 60.768 | 60.580 | 64.642 | 61.347 | 60.449 |
| | (26.27) *** | (25.48) *** | (29.39) *** | (26.83) *** | (23.39) *** |
| $R^2$ | 0.64 | 0.42 | 0.44 | 0.38 | 0.34 |
| $N$ | 182 | 162 | 65 | 155 | 124 |

Note: Average price for 1 March 2009–26 February 2012. Products with price below SEK 10/kg and three Fairtrade non-organic products in 250 g packages are excluded. Robust standard errors are used. * $p < 0.1$; ** $p < 0.05$; *** $p < 0.01$.

**Table A2.** Fairtrade coffee social premiums in SEK/kg (washed Arabica), inclusive of weight lost due to roasting (USD in parentheses).

| | Up to March 2011 | From April 2011 | Weighted Average |
|---|---|---|---|
| Fairtrade social premium | 1.85 (0.25) | 3.70 (0.50) | 3.11 (0.45) |
| Organic beans social premium | 3.70 (0.50) | 5.55 (0.80) | 4.66 (0.70) |

Note: Based on own calculations and Fairtrade Foundation (2012).

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
