# Peer review of "Fairtrade and Market Efficiency: Fairtrade-Labeled Coffee in the Swedish Coffee Market"

_economies, doi:10.3390/economies8020030_

Round 1
Reviewer 1 Report
In my view this article deals with an interesting topic which is announced in its title: "Fairtrade and Maket Efficiency". I would suggest, however, to add a subtitle so that the readers know in advance the field in which this important question will be analysed, for exemple "Fairtrade-labelled coffee in the Swedish market (2009-2012)".
The methodology looks accurate and the data provided in the study are robust enough to support the conclusions. Overall, I think that the results provide an advance in the current knowledge. I do not have any objections there.
In the conclusions, it would be desirable to go a little further on suggestions to improve the functioning of the Fairtrade system, as the only suggestion is based on another study (Griffiths, 2010), and it is described in a very broad manner. But I understand that this is not required for the acceptance of the article in an academic journal.
On the contrary, I think that it would be needed a little more description of types of coffee which are analysed in the article, at least for a reader that is not specialised in this field. For example, data in table 1 and table 2 are about conventional, Fairtrade organic, Fairtrade non organic, and organic not Fairtrade coffee. A non-expert reader in this field will probably assume that organic coffee is the one produced without any chemical fertilizers or pesticides. But maybe there are some specific conditions for Fairtrade labelling that would be necessary to know, and especially to understand the category "Fairtrade not organic". This would also be useful in order to attract a wider readership.
Author Response
Reviewer 1
Thanks for reviewing the paper and the nice comments.
In my view this article deals with an interesting topic which is announced in its title: "Fairtrade and Maket Efficiency". I would suggest, however, to add a subtitle so that the readers know in advance the field in which this important question will be analysed, for exemple "Fairtrade-labelled coffee in the Swedish market (2009-2012)".
I have changed the title, as suggested.
Fairtrade and Market Efficiency: Fairtrade-labelled Coffee in the Swedish Coffee Market
In the conclusions, it would be desirable to go a little further on suggestions to improve the functioning of the Fairtrade system, as the only suggestion is based on another study (Griffiths, 2010), and it is described in a very broad manner. But I understand that this is not required for the acceptance of the article in an academic journal.
I now dedicate one paragraph where I have refined the comments on how to improve the Fairtrade program.
“One suggestion is that rosters should offer two coffees that are identical in all respects except that one has a sum printed on it, which is donated to charity projects in poor countries (Griffiths, 2010). However, this would require monitoring by an external agency to be credible and it would remove the perceived link between consumption and coffee production with decent pay and working conditions. Moreover, the proposal to use donations is unlikely to have much policy relevance, since Fairtrade is a well-known organisation with a well-established label. In addition, consumers seem to be more willing to donate indirectly by paying a price premium on Fairtrade coffee than buying conventional coffee and giving a charity donation (Koppel and Schulze, 2013). A better strategy would be to improve the functioning of the Fairtrade system. One suggestion is that Fairtrade should require roasters to declare the difference between Fairtrade and conventional beans on packages. Although this would only give rough indication of how much of the retail price that goes to Fairtrade cooperatives, as prices vary, it would put a limit on how much roasters can charge for Fairtrade coffee. If unattractive to roasters in general, it could be adopted by those seriously engaged in Fairtrade, increasing their market shares and boosting competition in the coffee market.”
On the contrary, I think that it would be needed a little more description of types of coffee which are analysed in the article, at least for a reader that is not specialised in this field. For example, data in table 1 and table 2 are about conventional, Fairtrade organic, Fairtrade non organic, and organic not Fairtrade coffee. A non-expert reader in this field will probably assume that organic coffee is the one produced without any chemical fertilizers or pesticides. But maybe there are some specific conditions for Fairtrade labelling that would be necessary to know, and especially to understand the category "Fairtrade not organic". This would also be useful in order to attract a wider readership.
I have added a description on organic and Fairtrade coffee:
“Table 1 provides price information on the 188 ground coffee products available in packages of 250 g, 400–499 g and 500 g. There are 22 organic Fairtrade and 12 organic non-Fairtrade products. To be certified as organic, the coffee has to be grown without chemical pesticides or fertilizers, and be untreated with preservatives and other chemicals. For Fairtrade certification, producers have to comply with Fairtrade Standards, which aim to ensure the economic, social and environmental development of producers’ families, communities and organisations. The Fairtrade Standards include some environmental criteria, but the key feature is minimum prices and the premium paid on top of world markets prices. An additional premium is paid for organically certified coffee (Fairtrade Foundation, 2012).”
Reviewer 2 Report
The paper proposed an analysis of the distribution of the premium price paid by Swedish consumers for Fair trade coffe. Although the argument of the paper is interesting and the approach appropriate, the work need to be written in a more 'scientific sound' way. Author uses very often first person, which is something to avoid in scientific paper. English is not always good. Note that also the structure should be reorganized in classical sections (introduction, state of art, materials and methods, results and discussion, conclusions). Conclusions section should avoid refereces (which are rich in the discussion section), to focus more about what the study proof. Moreover, author should say why her/his study is worthy to be published: what is it new? Is it just a new application? Plus, it is maybe an idea to give the convertion SEK/dollar or SEK/euro: for not Swedish rearders it may be difficult to understand how much high Fairtrade prices are.
Author Response
Reviewer 2
Thanks for reviewing the paper and the useful comments.
Although the argument of the paper is interesting and the approach appropriate, the work need to be written in a more 'scientific sound' way. Author uses very often first person, which is something to avoid in scientific paper.
It is correct that the use of “I” is not permissible in many disciplines. However, it is quite common to write papers in first person in economics. I assume it is because there are many single-authored papers, and it is strange to use “we”. Moreover, when ‘we’ is used in a single-authored paper, “we” should include the reader, which many authors miss. On the other hand, not using any personal pronoun makes it difficult to avoid passive sentences.
Just to show that “I” is used when there is one author, I googled American Economic Review from 2020 and looked for single-authored articles. The first I found even starts the abstract with “I”
Ichihashi, S. (2020). Online privacy and information disclosure by consumers. American Economic Review, 110(2), 569-95.
Since I is used in top economics journals, I would prefer not to rewrite the paper removing all the “Is”.
English is not always good.
The paper has been copy-edited by a very experienced copy editor. She is American and lives in Sweden, i.e. the copy-editing was not done by a low-price company. Nevertheless, I have had the paper checked and a few sentences have been rewritten and some have been removed. I have also fixed some typos, which might have affect affected the English.
Note that also the structure should be reorganized in classical sections (introduction, state of art, materials and methods, results and discussion, conclusions). Conclusions section should avoid references (which are rich in the discussion section), to focus more about what the study proof.
I have followed the advice and written a Discussion and a Conclusion. I have also created a section called Results where the two former results sections, ‘How high are Fairtrade prices?’ and ‘Distribution of the premium paid for Fairtrade coffee’ are subsections. And there are no references in Conclusion now. In addition, I have removed the following paragraph from Introduction
“The Fairtrade label thus seems to create a coffee product that roasters and retailers can use to exploit their market power. Although the results are obtained with Swedish data, they are likely to also hold for consumer coffee markets in other Northern European countries, which have similar market structures. Whether they hold for other consumer coffee markets and for other products is an open question. It is possible that there is less scope for setting high markups on more homogenous products and on products with easily verifiable quality.“
Moreover, author should say why her/his study is worthy to be published: what is it new? Is it just a new application?
Thanks for the suggestion. I have added the following sentence Introduction
“As far as I know, apart from Podhorsky (2015), who assumes that the markup is the same for conventional and Fairtrade coffee, earlier studies have not attempted to estimate price-cost margins due to lack of information about costs (Samoggia and Riedel, 2018; Bissinger, 2019).”
I also conclude the review section with this
Thus, no study analyses a representative sample from a market where large companies produce Fairtrade coffees and grocery chains sell them. And no study attempts to estimate markups on Fairtrade coffee.
Plus, it is maybe an idea to give the convertion SEK/dollar or SEK/euro: for not Swedish rearders it may be difficult to understand how much high Fairtrade prices are.
I have added the US dollar values in the tables except the two that report regression results. I have also included US dollar values in parentheses in the text.
Thanks for the detailed suggestions about how to write Discussion and Conclusion. I have used the ones I think are relevant when re-writing the paper. I’ll certainly keep the list for future use. Please see Discussion and Conclusion for details.
Reviewer 3 Report
First of all let me state that I read the paper reviewed with a great pleasure, not only as a researcher, but also as a coffee consumer. I’d like to congratulate the Author (s) as the topic raised in the said paper is really very important nowadays. And probably it will be much more important in the future.
When reviewing scientific papers for publication, I usually start with a general overview in terms of a structure, abstract, literature review, methodology, findings of the research, discussion, conclusions, as well as limitations of the study.
The reviewed paper entitled “Fairtrade and Market Efficiency” is generally structured in a proper way. There are, however no sections ‘Discussion’, ‘limitations of the study’, and ’future directions of the research”. These sections should be added too, given this is a research paper.
The literature review is quite good and is strongly founded in the existing literature of the topic. Generally I claim that Author (s) provide solid theoretical foundations for the analysis using appropriate references. I would, however, recommend to add some references devoted to the latest literature associated with the topic in question (including Scopus Web of Science papers).
One should emphasize that the whole paper is very coherent and particular sub-parts fit together.
Additionally, one can see a smooth movement from one point to the another (end of deliberations in one sub-chapter creates also a beginning of a discussion in the next one).
The weak point of the article is the lack of the "Discussion" section. Please add this section in your article. Discussion is an interpretation of the results – implications, significance of results.
- Provide the response to the research question(s)
- Interpret results taking into account alternative explanations - where applicable
- What are the practical implications (and theoretical –where applicable) suggested by the results of your research.
- Include all limitations: this does not weaken your study, but adds to your credibility
- Future directions for research (incompletely answered questions) often derived from limitations.
- New questions which emerge from your research
- Be careful not to “go beyond” your data and results, in particular if the focus of your study is narrow
- You can “suggest”, or even “speculate” in the discussion, but it must be clearly evident what is derived from a result and what is your suggestion, comment or speculation, ...
- You may include a comparison with results of other similar/ compatible studies – if applicable.
Conclusion is the last part of the discussion or a separate chapter:
You may briefly summarize main results (if you haven‘t done this in the Discussion)
Bring the reader back to the research question – concluding with a larger and richer view of the problem/ question under investigation
Authors may add their own opinion and a broader comment of the results, their suggestions, recommendations, ...
Authors may add their proposals, suggestions, recommendations, evaluations, based on the results of the study - if appropriate as a separate chapter or subchapter.
Generally my opinion is positive. Though I have some minor remarks which may improve the paper
Author Response
Reviewr 3
Thanks for reviewing the paper and the nice comments.
First of all let me state that I read the paper reviewed with a great pleasure, not only as a researcher, but also as a coffee consumer. I’d like to congratulate the Author (s) as the topic raised in the said paper is really very important nowadays. And probably it will be much more important in the future.
When reviewing scientific papers for publication, I usually start with a general overview in terms of a structure, abstract, literature review, methodology, findings of the research, discussion, conclusions, as well as limitations of the study.
The reviewed paper entitled “Fairtrade and Market Efficiency” is generally structured in a proper way. There are, however no sections ‘Discussion’, ‘limitations of the study’, and ’future directions of the research”. These sections should be added too, given this is a research paper.
I have split up the old Conclusion into Discussion and Conclusion, and added some text.
The literature review is quite good and is strongly founded in the existing literature of the topic. Generally I claim that Author (s) provide solid theoretical foundations for the analysis using appropriate references. I would, however, recommend to add some references devoted to the latest literature associated with the topic in question (including Scopus Web of Science papers).
I have searched Scopus and Google Scholar and I found four relevant papers
These are
Bissinger K. 2019. Price Fairness: Two-Stage Comparison of Conventional and Fairtrade Prices, Journal of International Consumer Marketing, 31:2, 86-97,
Bissinger, K. and D. Leufkens, 2017. "Ethical food labels in consumer preferences", British Food Journal, Vol. 119 No. 8, pp. 1801-1814.
Samoggia, A., and Riedel B. 2018. Coffee consumption and purchasing behavior review: Insights for further research. Appetite, 129, 70-81.
Hejkrlik, Jiri, Jana Mazancova, and Karolina Forejtova. (2013). How effective is Fair Trade as a tool for the stabilization of agricultural commodity markets? Case of coffee in the Czech Republic. Agricultural Economics Czech, 59: 8-18.
One should emphasize that the whole paper is very coherent and particular sub-parts fit together.
Additionally, one can see a smooth movement from one point to the another (end of deliberations in one sub-chapter creates also a beginning of a discussion in the next one).
The weak point of the article is the lack of the "Discussion" section. Please add this section in your article. Discussion is an interpretation of the results – implications, significance of results.
- Provide the response to the research question(s)
- Interpret results taking into account alternative explanations - where applicable
- What are the practical implications (and theoretical –where applicable) suggested by the results of your research.
- Include all limitations: this does not weaken your study, but adds to your credibility
- Future directions for research (incompletely answered questions) often derived from limitations.
- New questions which emerge from your research
- Be careful not to “go beyond” your data and results, in particular if the focus of your study is narrow
- You can “suggest”, or even “speculate” in the discussion, but it must be clearly evident what is derived from a result and what is your suggestion, comment or speculation, ...
- You may include a comparison with results of other similar/ compatible studies – if applicable.
Conclusion is the last part of the discussion or a separate chapter:
You may briefly summarize main results (if you haven‘t done this in the Discussion)
Bring the reader back to the research question – concluding with a larger and richer view of the problem/ question under investigation
Authors may add their own opinion and a broader comment of the results, their suggestions, recommendations, ...
Authors may add their proposals, suggestions, recommendations, evaluations, based on the results of the study - if appropriate as a separate chapter or subchapter.
Generally my opinion is positive. Though I have some minor remarks which may improve the paper
Thanks for the detailed suggestions about how to write Discussion and Conclusion. I have used the ones I think are relevant when re-writing the paper. I’ll certainly keep the list for future use. Please see Discussion and Conclusion for details.
Round 2
Reviewer 2 Report
The author improve the paper, in particular the structure which was lack in the previous version. I still think that the use of the first person is to be avoid (singular or plural: does not matter!), or at least reduced in the paper. Editors may suggest or not to reduce it.